# The Gene *paaZ* of the Phenylacetic Acid (PAA) Catabolic Pathway Branching Point and *ech* outside the PAA Catabolon Gene Cluster Are Synergistically Involved in the Biosynthesis of the Iron Scavenger 7-Hydroxytropolone in *Pseudomonas donghuensis* HYS

**DOI:** 10.3390/ijms241612632

**Published:** 2023-08-10

**Authors:** Panning Wang, Yaqian Xiao, Donghao Gao, Yan Long, Zhixiong Xie

**Affiliations:** Hubei Key Laboratory of Cell Homeostasis, College of Life Sciences, Wuhan University, Wuhan 430072, China; 2015202040019@whu.edu.cn (P.W.); 2018102040012@whu.edu.cn (Y.X.); 2021202040036@whu.edu.cn (D.G.)

**Keywords:** *Pseudomonas donghuensis* HYS, 7-hydroxytropolone, biosynthetic pathway, *orf17–21* (*paaEDCBA*), *orf26* (*paaG*), phenylacetic acid metabolic pathway, *orf13* (*paaZ*), *ech*

## Abstract

The newly discovered iron scavenger 7-hydroxytropolone (7-HT) is secreted by *Pseudomonas donghuensis* HYS. In addition to possessing an iron-chelating ability, 7-HT has various other biological activities. However, 7-HT’s biosynthetic pathway remains unclear. This study was the first to report that the phenylacetic acid (PAA) catabolon genes in cluster 2 are involved in the biosynthesis of 7-HT and that two genes, *paaZ* (*orf13*) and *ech*, are synergistically involved in the biosynthesis of 7-HT in *P. donghuensis* HYS. Firstly, gene knockout and a sole carbon experiment indicated that the genes *orf17–21* (*paaEDCBA*) and *orf26* (*paaG*) were involved in the biosynthesis of 7-HT and participated in the PAA catabolon pathway in *P. donghuensis* HYS; these genes were arranged in gene cluster 2 in *P. donghuensis* HYS. Interestingly, ORF13 was a homologous protein of PaaZ, but *orf13* (*paaZ*) was not essential for the biosynthesis of 7-HT in *P. donghuensis* HYS. A genome-wide BLASTP search, including gene knockout, complemented assays, and site mutation, showed that the gene *ech* homologous to the ECH domain of *orf13* (*paaZ*) is essential for the biosynthesis of 7-HT. Three key conserved residues of *ech* (Asp39, His44, and Gly62) were identified in *P. donghuensis* HYS. Furthermore, *orf13* (*paaZ*) could not complement the role of *ech* in the production of 7-HT, and the single carbon experiment indicated that *paaZ* mainly participates in PAA catabolism. Overall, this study reveals a natural association between PAA catabolon and the biosynthesis of 7-HT in *P. donghuensis* HYS. These two genes have a synergistic effect and different functions: *paaZ* is mainly involved in the degradation of PAA, while *ech* is mainly related to the biosynthesis of 7-HT in *P. donghuensis* HYS. These findings complement our understanding of the mechanism of the biosynthesis of 7-HT in the genus *Pseudomonas*.

## 1. Introduction

*Pseudomonas* is a genus of ubiquitous Gram-negative γ-proteobacterium, the members of which are known for their capacity to colonize various ecological niches; in addition, they are highly versatile and adaptable [1,2,3]. This adaptability is considered to be associated with their diverse and highly elaborate siderophore systems [4,5,6,7]. Iron is an essential trace element for virtually all living organisms to maintain a normal life as it is involved in several key metabolic processes [8,9,10,11]. However, owing to its low solubility under physiological pH and aerobic conditions, iron is poorly bioavailable in the environment [12,13,14,15]. To survive in an iron-deficient environment, *Pseudomonas* can produce a variety of siderophores [16,17]. *P. donghuensis* HYS, isolated from Donghu Lake, was the type of strain of this newly classified species [18,19] and exhibited a high iron-chelating capacity [20,21,22]. *P. donghuensis* HYS *s*ecretes two types of iron scavengers: the fluorescent pyoverdine and a large quantity of nonfluorescent iron scavenger 7-hydroxytropolone (7-HT), which is a symmetrical seven-membered heteroatomic carbon ring containing a carbonyl group and two hydroxyl groups; 7-HT contributes to the notable iron-chelating ability in the culture supernatant of *P. donghuensis* HYS [20,21,22]. *P. donghuensis* HYS was more virulent to *C. elegans* than *P. aeruginosa* [23,24,25]. Other *P. donghuensis* strains also exhibited broad antibacterial activity, antifungal activity, and plant growth-promoting (PGP) activities [26,27,28,29], which are closely related to the 7-HT produced by these strains.

Siderophores are low-molecular-mass compounds (200–2000 Da) synthesized and secreted by bacteria to respond to iron deficiency. Siderophores possess a high affinity for iron (III) and can chelate the iron from different environments to ensure their survival [8,15,30]. The biosynthesis of siderophores is regulated by extracellular iron concentrations [31,32]. Among the variety of siderophores of Pseudomonads, pyoverdines (PVDs) are the primary siderophore and have been extensively studied due to their high affinity for Fe (III) and because they are widespread in *Pseudomonas* [33,34]. Further, *Pseudomonas* often produces a wide variety of secondary siderophores of lower affinity, such as pyochelin, pseudomonine, quinolobactin, and thioquinolobactin [16,35,36,37,38]. In addition to iron-chelating characteristics, secondary siderophores are often endowed with interesting characteristics, such as forming complexes with other metals, pathogenicity, antimicrobial activity, and biocontrol, which could well contribute to the competitiveness of these bacteria and better reflect the high adaptability of Pseudomonads to diverse environments and niche colonization [5,7,37,39]. The newly discovered nonfluorescent iron scavenger 7-hydroxytropolone (7-HT) is secreted in large quantities by the *P. donghuensis* HYS strain [20,22]. In addition, 7-HT is the main metabolite antagonistic to phytopathogenic fungi and plant–probiotic properties in other *P. donghuensis* strains [26,40].

The aerobic phenylacetic acid (PAA) catabolic pathway takes part in the biosynthesis of tropone/tropolone-related compounds, and the bifunctional fusion protein PaaZ plays a key role in catalyzing the degradation of PAA and the metabolic branching points for the biosynthesis of tropolones [41,42]. The fusion protein PaaZ consists of two protein domains: the C-terminal (R)-specific enoyl-CoA hydratase domain (PaaZ-ECH) and an N-terminal NADP + dependent aldehyde dehydrogenase domain (PaaZ-ALDH). When the oxidation function of the PaaZ-ALDH domain is deficient, the aerobic PAA catabolic pathway is truncated and branched to produce intermediate 2-hydroxycyclohepta-1,4,6-triene-1-formyl-CoA, which likely serves as the precursor for the biosynthesis of tropolonoids [42,43], such as tropodithietic acid (TDA) and roseobacticide in *Phaeobacter* inhibens [44,45], and 3,7-Dihydroxytropolone (DHT) isolated from *Streptomyces* species [43].

As identified in our previous work [20,22], to define the essential genes involved in the biosynthesis of the novel iron scavenger 7-HT, a random transposon insertion mutation was carried out in *P. donghuensis* HYS. The mutants that are deficient in siderophore yields were obtained by screening. The mutant genes were mainly located in two gene clusters, referred to as cluster 1 and cluster 2. Cluster 1 was composed of 12 genes related to the production of 7-HT. Among these genes was an *nfs* cluster (*orf6–orf9*) composed of four synthetases that participated in the synthesis of 7-HT. The deletion of these four genes eliminated 7-HT production. Furthermore, the production of 7-HT and the expression of *orf6–9* in relation to 7-HT biosynthesis was regulated by extracellular iron concentrations [20,22].

In *P. donghuensis* HYS, the biosynthetic pathway of 7-HT remains unclear, preventing its function from being fully applied and characterized. Thus, in this study, we focused on investigating the biosynthetic pathway of 7-HT. Whether these screened PAA genes *orf17–orf21* (*paaEDCBA*) and *orf26* (*paaG*) arranged in gene cluster 2 are actually involved in 7-HT biosynthesis, and the role of *paaZ* in the biosynthesis of 7-HT remain unclear. These questions urgently needs to be answered. Revealing the roles of PAA genes in 7-HT biosynthesis will not only provide further insight into the biosynthetic pathways of 7-HT, but will also more generally, improve understanding of the biosynthetic mechanisms for troplones and siderophores in the genus *Pseudomonas*.

## 2. Results

### 2.1. The Genes orf17–orf21 and orf26 in Cluster 2 Are Essential for the Production of the Iron Scavenger 7-HT

As shown in Appendix A, the transposon mutants corresponding to *orf17–21* and *orf26* produced significantly decreased total amounts of siderophores. The five genes related to the production of siderophores (*orf17*, *orf19–orf21*, and *orf26*) were identified in cluster 2 of *P. donghuensis* HYS (Appendix A and Figure 1A). Among these genes, *orf13–orf21* constituted a transcription operon. There was a cotranscriptional relationship between *orf26* and *orf27* (Appendix A).

To determine whether *orf17–21* and *orf26* affect the production of the iron scavenger 7-HT, we constructed an in-frame homologous deletion of mutants Δ*orf17*, Δ*orf18*, Δ*orf19*, Δ*orf20*, Δ*orf21*, Δ*orf26*, and Δ*orf17–21*, respectively. Each knockout strain showed markedly reduced siderophore production, both on CAS agar plates and in a liquid MKB medium, compared to that in the wild-type strain HYS (Figure 1B–D). On CAS agar plates, the yellow chelated halos of deleted mutants were much smaller than the WT strain HYS (Figure 1B), and the chelating ability of the liquid MkB culture supernatant of each knockout strain to the CAS detection solution was also remarkably lower (approximately one-sixth to seven times) than that of the WT strain HYS (Figure 1C). Moreover, the two characteristic absorption peaks of 7-HT, at 330 nm and 392 nm, disappeared, indicating a deficiency in the 7-HT yield. By contrast, only the characteristic absorption peak of pyoverdine, at 405 nm, could be detected in these mutants (Figure 1D). The results showed that the mutants no longer produced 7-HT. These results proved that the genes containing *orf17* to *orf21* and *orf26* in cluster 2 were indeed essential for the biosynthesis of 7-HT.

To determine whether the expression levels of these genes are inhibited by high extracellular iron concentrations, the transcriptional levels of these genes were detected by RTq-PCR during the exponential phase in *P. donghuensis* HYS inoculated in a liquid MKB medium and in MKB supplemented with 30 μM Fe^2+^, respectively. RTq-PCR results indicated that the transcriptional levels of the related genes *orf17–orf21* and *orf26* were significantly decreased in the high iron concentration-MKB medium (supplemented with 30 μM of FeSO_4_); these levels decreased by about 10-fold compared with the limited iron concentration-MKB medium (not supplemented with FeSO_4_), respectively (Figure 1E). It was confirmed that the expression levels of genes *orf17–orf21* and *orf26* were inhibited by high concentrations of iron. These results further indicated the production of HYS controls 7-HT by regulating the expression of *orf17–orf21*and *orf26* under different iron conditions and that these genes were also involved in the production of 7-HT.

Subsequently, a BLASTP search was performed with the protein sequences of these genes in NCBI. The results showed that these genes were related to the metabolism of PAA and shared high homologies to PAA cluster genes of *E. coli* K-12 (43–70%) (Appendix A). When cluster 2 of *P. donghuensis* HYS was aligned with the PAA cluster of *E. coli* K-12, we found that they were highly similar and showed that the *orf17*, *orf19–orf21*, and *orf26* genes, particularly, were homologous to *paaE*, *paaCBA*, and *paaG* PAA catabolon genes (Appendix A). From the above comparison results, it was preliminarily speculated that cluster 2 might be a PAA metabolic gene cluster.

The sole carbon utilization experiment showed that the *orf13*, *orf17–21*, and *orf26* genes in cluster 2 were involved in the catabolism of PAA in *P. donghuensis* HYS (Appendix A) and preliminary indicated that the PAA degradation pathway could be related to the biosynthesis of 7-HT in *P. donghuensis* HYS.

In brief, these results indicated that the *orf17–21* (*paaEDCBA*) and *orf26* (*paaG*) genes in cluster 2 were homologous to PAA metabolism genes and involved in the PAA catabolism in *P. donghuensis* HYS. Moreover, they were essential for the production of the iron scavenger 7-HT in *P. donghuensis* HYS.

ORF13 in cluster 2 of *P. donghuensis* HYS was homologous to PaaZ (Appendix A). It is unclear whether *orf13* (*paaZ*) is involved in the biosynthesis of 7-HT in *P. donghuensis* HYS.

### 2.2. ORF13, a Homologous Protein of PaaZ, Is Not Necessary for the Biosynthesis of 7-HT in P. donghuensis HYS

To investigate whether *orf13* (*paaZ*) also participated in the biosynthesis of 7-HT, the *orf13* gene was deleted in HYS. Then, the ability of the Δ*orf13* strain to produce siderophores, especially the iron scavenger 7-HT, was determined.

On CAS agar plates (under normal or UV light), the yellow chelated halo of the Δ*orf13* strain was not significantly different from that of the WT HYS strain (Figure 1B). This phenomenon preliminarily indicated that the deletion of *orf13* had no obvious effect on the total amount of siderophores. The yield of siderophores on the Δ*orf13* strain in the liquid MKB medium supernatant was reduced only by about one-fifth to one-sixth compared to that of the wild-type HYS strain (Figure 1C). Similarly, the characteristic absorption peaks of 7-HT in the Δ*orf13* strain, at 330 nm and 392 nm, were only slightly decreased compared with the WT strain HYS (Figure 1D). The deletion of *orf13* could not completely eliminate the synthesis of 7-HT, but interestingly, the RTq-PCR result indicated that the transcriptional level of *orf13* was inhibited by high-concentration iron (Figure 1E).

The results showed that *orf13* had no significant effect on the yield of fluorescent pyoverdine and nonfluorescent iron scavenger 7-HT in HYS. The sole carbon experiment showed that *orf13* was involved in the catabolism of PAA in *P. donghuensis* HYS (Appendix A).

Through amino acid sequence alignment, it was found that ORF13 (PaaZ) also consisted of two domains, the N-terminal PaaZ-ALDH and the C-terminal PaaZ-ECH domain, and that the active site of the ALDH domain of PaaZ in *P. donghuensis* HYS was 258 Glu (E). To inactivate the ALDH domain, the site was mutated to Gln (Q) (E258Q). To further investigate the relationship between *orf13* (*paaZ*), and the production of 7-HT, we constructed derived strains of the pBBR2-*paaZ* and site-mutated pBBR2-*paaZ*E258Q in the HYS strain. The changes in the 7-HT production of *paaZ* and *paaZ*E258Q overexpressing strains were observed in a liquid MKB medium (Figure 2 and Figure 3B).

Compared with the HYS strain, the characteristic absorption peaks of 7-HT at 330 nm and 392 nm were only slightly reduced in the Δ*paaZ* strain (Figure 1D), while, interestingly, in the *paaZ* overexpressed strain HYS/pBBR2-*paaZ*, the characteristic absorption peaks of 7-HT disappeared, and only the absorption peak of pyoverdine at 405 nm remained compared with the empty vector strain HYS/pBBR2, indicating that the production of 7-HT was eliminated (Figure 2). In the *paaZ*E258Q-overexpressed strain HYS/pBBR2-*paaZ*E258Q, the characteristic absorption peaks of 7-HT appeared again compared with the HYS/pBBR2-*paaZ* strain, indicating that 7-HT was produced again (Figure 2).

These data show that the overexpression of *paaZ* in the wild-type HYS strain inhibited the production of 7-HT, and compared with the overexpression of *paaZ*, the overexpression of *paaZ*E258Q restored the production of 7-HT. Furthermore, these results further indicated that ORF13 (PaaZ) did not play a key role in the biosynthesis of 7-HT in *P. donghuensis* HYS.

In summary, although ORF13 is a homologous protein of PaaZ, interestingly, Δ*orf13* (Δ*paaZ*) could not completely eliminate the production of 7-HT, indicating that *orf13* is not essential for the production of 7-HT. Thus, it is speculated that there may be at least one other protein homologous to ORF13 (PaaZ) or a protein homologous to the PaaZ-ECH domain in *P. donghuensis* HYS, which could be involved in the biosynthesis of 7-HT.

### 2.3. The Gene Ech Is Homologous to the C-Terminal ECH Domain of ORF13

To obtain predicted ORF13 (PaaZ) homologous proteins or PaaZ-ECH-like proteins, we performed a genome-wide BLASTP search in *P. donghuensis* HYS using the whole ORF13 amino acid sequences or its C-terminal ECH domain as a query. Using the entire ORF13 protein sequence for BLASTP, eight homologous sequences were obtained in *P. donghuensis* HYS (Table 1, Figure 3A), six of which were aldehyde dehydrogenase family proteins (Sequence Number: 1, 3, 5, 6, 7, 8) homologous to the N-terminal PaaZ-ALDH domain. The other two were MaoC family dehydratases (Sequence Number: 2, 4)—(R)-hydratase [(R)-specific enoyl-CoA hydratase] and NodN (nodulation factor N)—homologous to the C-terminal PaaZ-ECH domain (Table 1, Figure 3A). Consequently, based on the above comparative analysis, the gene UW3_RS0113785 was renamed *ech*, and UW3_RS0112810 was renamed *NodN*. Using the C-terminal ECH domain amino acid sequences of ORF13 as a query (Appendix A), these two hydratase proteins (Sequence Number: 2, 4) were also obtained.

To identify whether the two hydratases, *ech*, and *NodN*, were involved in the biosynthesis of 7-HT (Table 1, Figure 3A), the Δ*ech* and Δ*NodN* knockout strain was constructed. Compared with the WT HYS strain and in the Δ*ech* strain, the two characteristic absorption peaks of 7-HT, at 330 nm and 392 nm, disappeared, indicating that the knockout of *ech* eliminated the production of 7-HT. However, in the Δ*NodN* strain, the two characteristic absorption peaks of 7-HT did not significantly change compared with HYS, indicating that *NodN* did not participate in the biosynthesis of 7-HT (Figure 3C). The results suggest that *ech*, but not *NodN*, was involved in the synthesis of 7-HT.

### 2.4. ech Is Essential for the Biosynthesis of the Iron Scavenger 7-HT

To identify whether *ech* was related to the biosynthesis of the iron scavenger 7-HT in *P. donghuensis* HYS, *ech* was knocked out in wild-type HYS and Δ*paaZ*. Then, the ability of Δ*ech* and Δ*paaZ*Δ*ech* strains to produce 7-HT was tested in liquid MKB (Figure 4).

Compared with the wild-type HYS, in the Δ*ech* mutant strain, the two characteristic absorption peaks of 7-HT disappeared, and only the absorption peak of pyoverdine was left (Figure 4A). Similarly, in the Δ*paaZ*Δ*ech* mutant strain, the characteristic absorption peaks of 7-HT disappeared, and only the absorption peak of pyoverdine was detected compared with the Δ*paaZ* strain (Figure 4A). The data confirmed that *ech* was essential for the production of the iron scavenger 7-HT.

To further explore the relationship between *ech* and the production of 7-HT, the complemented strains Δ*ech*/pBBR2-*ech*, Δ*paaZ*Δ*ech*/pBBR2-*ech*, Δ*ech*/pBBR2, and Δ*paaZ*Δ*ech*/pBBR2 were constructed (Figure 4B,C). The absorption spectrum of the complemented strains in liquid MKB culture supernatants was measured, and the characteristic absorption peaks of 7-HT were recovered in the Δ*ech*/pBBR2-*ech*-complemented strain compared with those of the Δ*ech*/pBBR2 strain (Figure 4B). The Δ*paaZ*Δ*ech*/pBBR2-*ech*-complemented strain exhibited a phenotype similar to that of the Δ*ech*/pBBR2-*ech* strain, which could also restore the production of 7-HT compared with the Δ*paaZ*Δ*ech*/pBBR2 strain (Figure 4C).

The above data show that the complement of *ech* could restore the ability of the knockout strains to produce 7-HT, which further indicated that *ech* was necessary for the biosynthesis of 7-HT and played a crucial role in the biosynthesis of 7-HT in *P. donghuensis* HYS.

### 2.5. Identification of Key Conserved Residues of Ech in P. donghuensis HYS

To further understand the effect of the gene *ech* on the biosynthesis of iron-scavenging 7-HT in *P. donghuensis* HYS, we performed the sequence alignment of ECH with those of other 12 paaZ-ECH domains, including *E. coli* K12. It was found that the conserved catalytic residues of ECH were Asp(D)-39, His(H)-44, and Gly(G)-62 (Figure 3B and Appendix A), which is similar to the three conserved catalytic residues of the PaaZ-ECH domain in *E.coli* K12 (Asp-561, His-566, and Gly-584) [42]. The 3D (three-dimensional) spatial protein structure of ECH in *P. donghuensis* HYS was predicted. The key residues Asp39, His44, and Gly62 are represented as stick models; the location of the active sites of the enzyme is indicated (Figure 5B).

By mutating these assumed key conserved sites (D39, H44, and G62) of the ECH domain into Ala (A), respectively, the complementary vectors pBBR2-*ech*D39A, pBBR2-*ech* H44A, and pBBR2-*ech*G62A were constructed and introduced into the Δ*ech* strain to obtain the site mutant complementary strains Δ*ech*/pBBR2-*ech*D39A, Δ*ech*/pBBR2-*ech*H44A, and Δ*ech*/pBBR2-*ech*G62A, respectively.

Compared with the strains Δ*ech*/pBBR2-*ech* and HYS, the characteristic absorption peaks of 7-HT (at 330 nm and 392 nm) were not restored in these site mutant complementary strains (Δ*ech*/pBBR2-*ech*D39A, Δ*ech*/pBBR2-*ech*H44A, and Δ*ech* /pBBR2-*ech*G62A (Figure 5A)), while nonmutant *ech* complementary strains could resume the production of 7-HT.

The results indicated that, due to the site mutations of the key conserved catalytic sites, *ech* was inactivated; thus, the production of 7-HT could not be restored. The results also preliminarily identified that these three residues (Asp39, His44, and Gly62) were the key conserved catalytic sites of *ech* and further proved that *ech* was necessary for the biosynthesis of 7-HT in *P. donghuensis* HYS. At present, there is no report describing the role of *ech* involved in 7-HT biosynthesis.

### 2.6. The Gene paaZ Cannot Complement the Role of Ech in the Production of 7-HT in P. donghuensis HYS

To further explore the relationship between genes *ech* and *paaZ* and the production of 7-HT, pBBR2-*paaZ*, and site-mutated pBBR2-*paaZ*E258Q were introduced into Δ*ech* strains to observe whether the production of 7-HT was recovered.

Compared with HYS, the characteristic absorption peaks of 7-HT, at 330 nm and 392 nm, were not recovered in the Δ*ech*/pBBR2-*paaZ* and Δ*ech*/pBBR2-*paaZE258Q* strain, indicating that 7-HT production was not recovered (Figure 6A). The transcriptional levels of *paaZ* in Δ*ech*/pBBR2-*paaZ* and Δ*ech*/pBBR2-*paaZE258Q* were approximately 180 and 160 times higher than those in wild-type HYS, respectively, indicating the successful expression of *paaZ* and *paaZE258Q* (Figure 6B).

These results indicated that the introduction of pBBR2-*paaZ* or pBBR2-*paaZ*E258Q plasmid into Δ*ech* could not restore the 7-HT production of Δ*ech*, suggesting that *paaZ* and *paaZE258Q* (the ECH domain of PaaZ) could not complement the role of *ech* in 7-HT biosynthesis. Furthermore, it was demonstrated that *ech* plays a more important role in the production of 7-HT than *paaZ*, which is consistent with the phenomenon whereby Δ*paaZ* (Δ*orf13*) affects the production of 7-HT only slightly (Figure 1). In addition, it was demonstrated that *PaaZ* was involved in 7-HT biosynthesis to a small extent and could not complement the disappearance of 7-HT production due to the deletion of *ech*.

### 2.7. PaaZ Is Mainly Involved in the PAA Catabolic Pathway, While Ech Is Not

In *P. donghuensis* HYS, *paaZ* is a gene in the PAA metabolism gene cluster 2, and *ech* is an R-type enoyl-CoA with homology to the PaaZ-ECH domain outside gene cluster 2. This preliminary shows that *paaZ* is related to PAA degradation (Appendix A); thus, to investigate whether *ech* is involved in PAA metabolism, the ability of *ech* to utilize PAA was tested by adding PAA to an M9 minimal medium as a single carbon source.

Compared with the wild-type HYS strain, there was no significant growth difference between the Δ*paaZ*, Δ*ech*, and ∆*paaZ*Δ*ech* strains with glucose as the single carbon source, indicating that the deletion of these genes did not affect the glucose utilization of *P. donghuensis* HYS (Figure 7A). When PAA was used as the sole carbon source, compared with the wild-type HYS, the two knockout strains, ∆*paaZ* and ∆*paaZ*∆*ech*, showed a growth defect, but there was no growth defect of the ∆*ech* strain (Figure 7B), indicating that the Δ*paaZ* strain could not utilize PAA and that the deletion of *ech* did not affect the utilization of PAA. Thus, it could be concluded that *paaZ* in gene cluster 2 was involved in PAA catabolism, while *ech* was not involved in PAA metabolism in *P. donghuensis* HYS.

In summary, *paaZ* in gene cluster 2 was mainly related to PAA catabolism, while *ech* was not involved in PAA metabolism in *P. donghuensis* HYS, and *ech* was mainly involved in the biosynthesis of 7-HT in *P. donghuensis* HYS. These results confirm that *ech* and *paaZ* have functional differentiation roles in *P. donghuensis* HYS, which should direct further studies on the biosynthesis mechanism of 7-HT.

### 2.8. The Expression Differences of paaZ and ech and the Response to Extracellular Iron Concentration in P. donghuensis HYS

The genes *paaZ* and *ech* have different roles in the biosynthesis of 7-HT. *ech* is mainly involved in the biosynthesis of 7-HT, while *paaZ* is mainly related to PAA catabolism in *P. donghuensis* HYS. Thus, to investigate whether there were differences in the expression levels of *paaZ* and *ech* during the period of 7-HT mass synthesis in *P. donghuensis*, during the experiment, the HYS strain was inoculated in the MKB medium and cultivated to the exponential phase (8 h), which was the period when a large amount of 7-HT was synthesized in HYS [20,21].

It was found that the gene expression level of *ech* was about 19 times that of *paaZ* (Figure 8A), indicating that *paaZ* and *ech* had different expression levels in the exponential period when 7-HT was produced in large quantities. This is consistent with the phenomenon that Δ*ech* does not produce 7-HT and has a greater impact on the production of 7-HT, while Δ*paaZ* has little effect on the production of 7-HT.

To explore the relationship between the transcription level of *ech*, *paaZ*, and extracellular iron, the HYS strain was cultured to the logarithmic phase in the MKB medium and MKB medium supplemented with 30 μM FeSO_4_, respectively. The transcription levels of *paaZ* and *ech* were detected. Compared to the low-iron environment (iron-limited MKB medium), in the high-iron environment (MKB supplemented with 30 μM FeSO_4_), the transcription levels of *paaZ* and *ech* were somewhat reduced, and the transcription level of *paaZ* decreased by about 10 times; the transcription level of *ech* dropped by about 2 times (Figure 8B). The above results indicate that the transcription levels of *paaZ* and *ech* were inhibited by a high iron concentration, and the inhibition degree of iron to *ech* was weaker than that of *paaZ*.

## 3. Discussion

The novel iron scavenger 7-hydroxytropolone (7-HT) is secreted by *P. donghuensis* HYS in large quantities and contributes to the notable iron-chelating activity of *P. donghuaensis* HYS [20,21,22]. Further, 7-HT contributes to the pathogenicity of *C. elegans* in *P. donghuensis* HYS [23,24,25]. In other *P. donghuensis* strains, 7-HT is also the main metabolite to antifungal and plant protective agent [26,40]. However, its biosynthetic pathway remains unclear. In this study, based on gene knockout, and complementation experiments, we identified that the *orf17–21* (*paaEDCBA*) and *orf26* (*paaG*) genes in gene cluster 2 were essential for the production of 7-HT in *P. donghuensis* HYS. A sole carbon source experiment indicated that o*rf13* (*paaZ*), *orf17–21* (*paaEDCBA*), and *orf26* (*paaG*) related to the biosynthesis of 7-HT were involved in the PAA catabolon pathway. The data indicated that the PAA catabolon pathway participated in the production of 7-HT in *P. donghuensis* HYS.

This mechanism is similar to the biosynthesis of TDA and roseobacticide in *P. inhibens* [44,45] and the biosynthesis of 3,7-Dihydroxytropolone (DHT) in *Streptomyces* spp. [43]. That is, the PAA metabolic pathway provides the seven-membered ring carbon skeleton of these compounds.

Interestingly, although ORF13 was homologous to PaaZ, *orf13* (*paaZ*) was not essential for the production of 7-HT in *P. donghuensis* HYS. The gene *ech* outside of cluster 2 was homologous to the C-terminal PaaZ-ECH domain, and *ech* was involved in the biosynthesis of 7-HT in *P. donghuensis* HYS. These results confirm that *ech* and *paaZ* have functional differentiation roles and a synergistic effect in *P. donghuensis* HYS; *ech* is mainly related to the biosynthesis of 7-HT and *paaZ* mainly plays a role in PAA catabolism in *P. donghuensis* HYS. At present, there is no report describing the role of *ech* in 7-HT biosynthesis. The gene *paaZ* was found to participate in the PAA metabolic branching point [41,42]. Although *paaZ* was also found to participate in the PAA catabolism in *P. donghuensis* HYS, it was not necessary for the biosynthesis of 7-HT, and the gene *ech* located outside cluster 2 is involved in 7-HT biosynthesis—an unexpected finding.

A new feature was observed in *P. donghuensis* HYS: there was a complete PAA degradation in cluster 2 and an independent *ech* outside the PAA cluster, and *paaZ* had complete ALDH and ECH functions, which is different from the feature observed in the production of TDA and roseobacticide in *P. inhibens* [44,45] and the biosynthesis of 3,7-Dihydroxytropolone (DHT) in *Streptomyces* spp. [43]. In *P. inhibens*, in addition to the *paaZ1* of the PAA catabolon gene cluster, a specific didomain enzyme, *PaaZ2*, outside the PAA gene cluster was found. The ECH domain of *PaaZ2* is highly homologous to *PaaZ1* in *P. inhibens*, while the N-terminal ALDH domain shows very low homology. In *Streptomyces*, the *trlA* gene, located outside the paa gene cluster, is involved in the production of 3,7-Dihydroxytropolone (DHT) [43]. The *trlA* gene encodes a single-domain (ECH) protein, homologous to the C-terminal ECH domain of *E. coli* bifunctional protein PaaZ. TrlA truncates the PAA catabolic pathway and redirects it toward the formation of DHT [43]. The differences are that in *P. inhibens*, there are two *paaZs*: *paaZ1* and *paaZ2*, while in *Streptomyces*, there is only one TrlA homologous to the paaZ-ECH domain without PaaZ. In HYS, there is a *paaZ* and an *ech* which are synergistically involved in the synthesis of 7-HT and metabolism of PAA.

It was speculated that the coexistence of *paaZ* (*orf13*) and *ech* would rapidly produce more 7-HT to cope with environmental stress and save energy, giving the strain a competitive advantage and ensuring its survival. The *paaZ* and *ech* could cooperate to allow *P. donghuensis* HYS to adapt to challenging environmental conditions. *paaZ* and *ech* exist simultaneously and are functionally differentiated, which not only ensures the metabolism of PAA but also ensures the synthesis of 7-HT in large quantities when needed in order to obtain competitive advantages and realize the regulation of the 7-HT synthesis pathway.

The two genes, *paaZ*, and *ech*, had different expression levels in the exponential period. The expression level of *ech* was significantly higher than that of *paaZ* in *P. donghuensis* HYS. This is consistent with the phenomenon that Δ*ech* does not produce 7-HT and has a greater impact on the production of 7-HT, while Δ*paaZ* has little effect on the production of 7-HT. In addition, it is interesting that iron has a stronger inhibitory effect on *paaZ* and a weaker inhibitory effect on the transcription of *ech*. We speculated that the reason for this may be that *ech* is also involved in other reaction pathways not strictly regulated by iron concentration. Furthermore, 7-HT can also act as an antibiotic [40,46], enhancing the competitive fitness of bacteria against other microorganisms. Therefore, it is speculated that when the external iron condition is not so scarce, *ech* and its expression at a high level guarantee the mass synthesis of 7-HT, and *ech* can be synthesized both rapidly and earlier by HYS in order to chelate iron and restrict the growth of competitors, which gives *P. donghuensis* HYS a competitive advantage. This could provide insight into how *P. donghuensis* HYS adjusts the expression of its *paaZ* and *ech* genes—key branching points in the 7-HT biosynthesis pathway—to match varying levels of iron, competition, and environmental pressure.

## 4. Materials and Methods

### 4.1. Bacterial Strains, Plasmids, and Growth Conditions

The bacterial strains and plasmids used in this work are listed in Appendix A. *P. donghuensis* HYS isolated from Donghu Lake was used as the wild-type strain. *Escherichia coli* S17-1 λpir [47] was used as the host for DNA manipulation by conjugating with *Pseudomonas* strains. Plasmid pEX18Gm [48] was used as a vector for gene knockout; plasmid pBBR1MCS-2 [49] was used for gene overexpression, and the promoterless *lacZ* reporter plasmid pBBR5Z [20] was used for promoter assays. *E. coli* strains were routinely cultured in a Luria–Bertani (LB) medium at 37 °C. *P. donghuensis* HYS and its derivative strains were grown in an LB medium and iron-deficient MKB medium (15 mL/L glycerol, 2.5 g/L K_2_HPO_4_, pH 7.2 subsequently supplemented with 2.5 g/L MgSO_4_ and 5 g/L casamino acid) at 30 °C. When required, a final concentration of 30 µM FeSO_4_ was added to the MKB medium. When necessary, antibiotics were added at the following final concentrations: for *E. coli* strains, 50 μg/mL kanamycin and 10 μg/mL gentamicin; for *P. donghuensis* HYS and its derivative strains, 25 μg/mL chloramphenicol, 50 μg/mL gentamicin, and 50 μg/mL kanamycin were used.

### 4.2. Construction of In-Frame Deletion Mutants and Complement Strains of P. donghuensis HYS 

Routine molecular genetic manipulation, including PCR, agarose gel electrophoresis, gel extraction, restriction enzyme digestion, and transformation, were performed using standard procedures [50]. The DNA amplification primers used are listed in Appendix A.

The deletion was performed by the double homologous recombination method [48]. To construct the deletion plasmids, fragments containing 400–600 bp regions upstream and downstream of the target genes and several nucleotides of the ORFs were amplified and ligated into the suicide vector pEX18Gm. The correct recombinant plasmids were verified by sequencing and then transferred into HYS and its derivative strains via conjugation from *E. coli* S17-1 (λ pir). Single recombinants were selected on LB agar plates with chloramphenicol and gentamicin, which were based on their resistance to two antibiotics. These recombinants were further incubated overnight at 30 °C for at least 24 h in 5 mL of the liquid LB medium without antibiotics to produce a second allelic exchange. The appropriate strain culture was then diluted by gradient on LB agar plates with 5% (wt/vol) sucrose. The correct deletion mutants were selected and further confirmed by PCR and DNA sequencing.

The plasmids used for complementation and overexpression were constructed by cloning the Shine–Dalgarno (SD) sequences and open reading frames (ORFs) of target genes into the broad-host-range vector pBBR1MCS-2. The plasmid with the target gene fragment was transferred into *E. coli* S17-1 (λ pir), and then the *E. coli* S17-1 (λ pir) carrying recombinant plasmids and containing the correct fragments was conjugated with *P. donghuensis* HYS mutant strains. After conjugation with the corresponding mutants, correct monoclonal targets were selected with double-antibiotic treatments, plasmid extraction, and enzyme digestion verification.

### 4.3. Construction of Point Mutations in P. donghuensis HYS

The plasmids used for point mutation were constructed by cloning target genes with the point mutation site into the broad-host-range vector pBBR1MCS-2. An overlap PCR was used to amplify target genes containing the point mutation site, and other procedures were similar to those of the gene complementation in this study. Oligonucleotide primers were used to amplify target genes with the point mutation sites shown in Appendix A.

### 4.4. Siderophore Determination Assays

The production of siderophores was determined using the following methods: on universal Chrome Azurol S (CAS) blue agar plates [51], in which the MM9 solution was replaced by a pH 6.8 phosphate buffer [22], siderophores were measured using their high affinity for iron (III). For the siderophores secreted by the bacteria chelate iron from the medium, their color turns from blue to orange. The orange halos around the colonies indicated the secretion of siderophores [51]. In this study, the strains were inoculated in the MKB medium and incubated at 30 °C for 24 h. Then, 10 µL of each MKB culture (the optical density at 600 nm (OD_600_) was adjusted to 1.0) was dropped onto a CAS agar plate, and the quantities of siderophores were measured by the chelated halos formed after incubating for 24 h at 30 °C.

For the liquid MKB medium, the appropriate dilution of each filtered supernatant was mixed with an equal volume of the CAS assay solution [51] using double-distilled water (ddH_2_O) as a reference. After 1 h of incubation at room temperature, the absorbances of the samples (As) and reference (Ar) at 630 nm were detected. Siderophore units were calculated by subtracting the sample absorbance values from the reference according to the following equation [52]: 100 × (*Ar* − *As*)/*Ar* = % siderophore units.

To obtain the characteristic absorption peaks of the siderophores in a liquid MKB medium, the filtered supernatants of 24 h MKB cultures were normalized to an OD_600_ of 0.5. The characteristic absorption peaks of 7-HT were at approximately 330 and 392 nm, and that of pyoverdine was at approximately 405 nm. Then, the absorption spectra were measured every 0.5 nm using a UV/visible spectrophotometer (UV-2550; Shimadzu, Kyoto, Japan).

### 4.5. Real-Time qPCR

Triplicate PCR reactions were carried out according to the manufacturer’s instructions (SYBR Premix Ex Taq Tli RNaseH Plus; TaKaRa, Kusatsu, Japan) in a qPCR machine (CFX96 real-time PCR detection system; Bio-Rad Technologies, Hercules, CA, USA). The reaction system and cycling conditions for qPCR were performed according to the manufacturer’s protocol (SYBR Premix Ex Taq Tli RNaseH Plus; TaKaRa). The experimental data were analyzed using Bio-Rad CFX Manager 3.1 software (Bio-Rad Technologies). To quantify relative gene expression, real-time qPCR data were analyzed by the 2^−ΔΔCT^ method [53]. The relative expression of the target genes is shown as the ratio of samples to the wild type after normalization to the reference gene *rpoB* [54,55].

### 4.6. Bioinformatic Analysis

A genome-wide BLASTP search was performed for predicted ORF13 (PaaZ) homologous proteins or PaaZ-ECH-like proteins using the whole ORF13 amino acid sequence or its C-terminal ECH domain as a query in NCBI BLASTP. Standard databases were searched using the following queries: non-redundant (nr) protein sequences, organism optional, and *Pseudomonas donghuensis* (Taxid: 1163398). Algorithm parameters used the default settings (Expect threshold: 0.05).

The sequences in Appendix A were obtained from NCBI and compared using the clusterW method by MEGA 11 version 11.0.13 software; some sequences refer to Nelson L. Brock et al. [45].

### 4.7. RNA Isolation and Reverse Transcription (RT)-PCR

A total of 1 mL of the culture of *P. donghuensis* HYS and its derivative strains in the exponential phase (at an OD_600_ of 0.6 to 0.8) was harvested in 50 mL of a liquid MKB medium or medium supplemented with 30 μM FeSO_4_·7H_2_O incubating at 30 °C. Then, the supernatant was removed by centrifugation (13,000× *g*, 1 min) at 4 °C. The total RNA was extracted using the TRIzol reagent (Ambion, Invitrogen, Carlsbad, CA, USA) according to the manufacturer’s instructions. Then, genomic DNA was removed according to the method proposed by Chen et al. [22].

RNA was quantified using a NanoDrop 2000 (Thermo Fisher Scientific, Shanghai, China) and was reverse-transcribed to generate cDNA using the PrimeScript RT Reagent Kit with gDNA Eraser (TaKaRa; Kyoto, Japan). The RNA was stored at −80 °C, and the cDNA was stored at −20 °C. cDNA was used as a template for real-time qPCR.

### 4.8. Accession Numbers

The GenBank accession numbers for the genes *orf17* (*paaE*), *orf18* (*paaD*), *orf19* (*paaC*), Δ*orf20* (Δ*paaB*), *orf21* (*paaA*), *orf26* (*paaG*), *orf13* (*paaZ*), *ech*, and *NodN* from *P. donghuensis* HYS are UW3_RS0120700, UW3_RS0120705, UW3_RS0120710, UW3_RS0120715, UW3_RS0120720, UW3_RS0120745, UW3_RS0120680, UW3_RS0113785, and UW3_RS0112810, respectively.

## 5. Conclusions

In summary, this study is the first to report that the *paaABCDE* and *paaG* genes in cluster 2 are involved in the biosynthesis of 7-HT and that two genes (*paaZ (*(*orf13*)) at the branching point of the PAA metabolic pathway and *ech*) are synergistically involved in the biosynthesis of 7-HT in the genus *Pseudomonas*. The gene *paaZ* mainly plays a role in the degradation of PAA and slightly affects the biosynthesis of 7-HT, while *ech* mainly participated in the biosynthesis of 7-HT in *P. donghuensis* HYS. Moreover, *ech* and *paaZ* were distributed far apart in the genome of HYS, and *ech* was encoded outside of the PAA metabolic gene cluster 2, where *paaZ* was found, which should direct further studies on the biosynthesis mechanism for 7-HT. This study revealed a natural association between the catabolism of PAA and the biosynthesis of 7-HT: genes related to PAA metabolism are involved in the biosynthesis of the siderophore 7-HT in *P. donghuensis* HYS. This association has not been reported previously in the genus *Pseudomonas*. This study complements our understanding of the mechanism of the biosynthesis pathway of 7-HT and provides directions for the study of the 7-HT biosynthesis pathway in *P. donghuensis* HYS. Moreover, this study also enriched research on the biosynthesis of troplones and siderophores in the genus *Pseudomonas*.

## Figures and Tables

**Figure 1 ijms-24-12632-f001:**
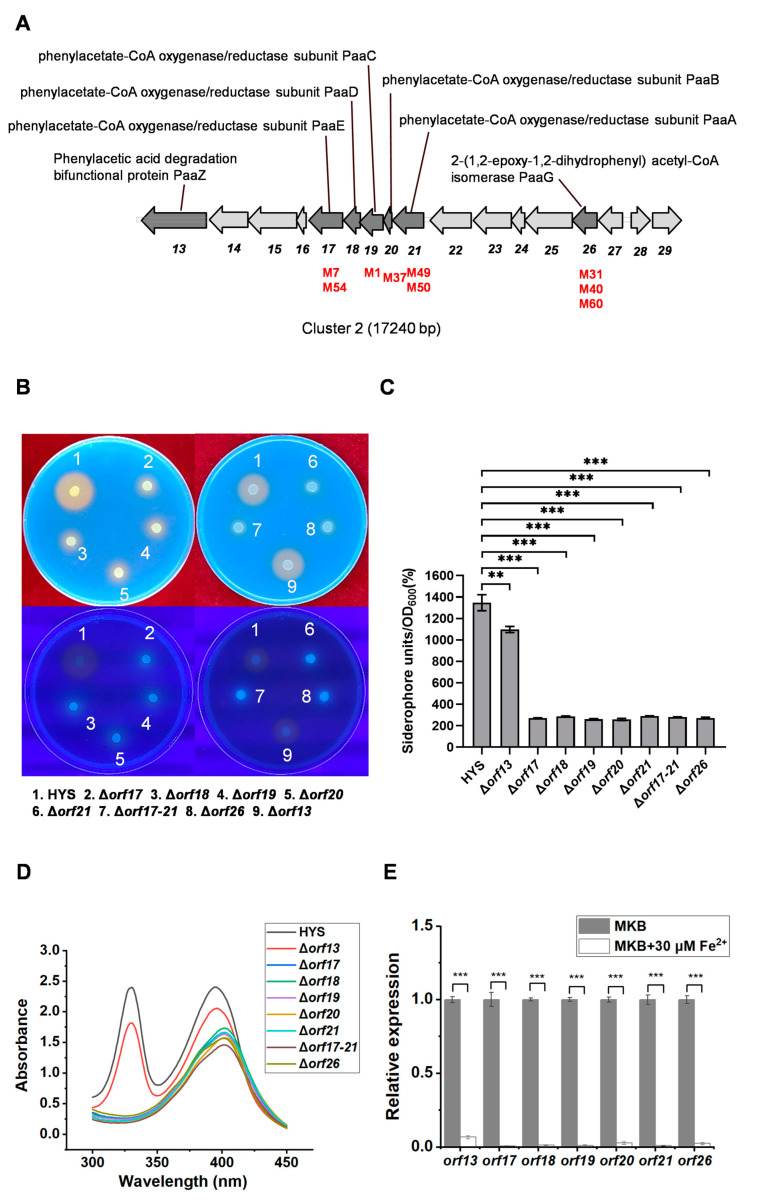
Effects of *orf13*, *orf17–orf21*, and *orf26* on siderophore production in *P. donghuensis* HYS. (**A**) Diagram of the gene arrangement of cluster 2 of *P. donghuensis* HYS. The arrows and their directions indicate the locations and direction, respectively, of the transcription of the predicted genes in cluster 2. The red font indicates the transposon mutant strains and their insertion positions. (**B**) Siderophore production in wild-type HYS and the Δ*orf13*, Δ*orf17*, Δ*orf18*, Δ*orf19*, Δ*orf20*, Δ*orf21*, Δ*orf17–21*, and Δ*orf26* mutants were tested on CAS agar plates under normal light (upper) or UV light (lower). (**C**) Siderophore production in wild-type HYS and the derivative mutant strains in liquid MKB medium was determined as siderophore units (percent) using the CAS liquid assay. (**D**) Absorption spectra of the supernatants of 24 h liquid MKB cultures from wild-type HYS and derivative strains. (**E**) The relative transcription level of *orf13*, *orf17–orf21*, and *orf26* in *P. donghuensis* HYS in high and low extracellular iron concentrations. RNA was isolated from the indicated strains grown at an exponential phase at 30 °C in a liquid MKB culture supplemented with or without 30 µM FeSO_4_·7H_2_O. Each value is the average from three different cultures ± the standard deviation. The error bars indicate standard deviations from 3 replicates (*n* = 3). ** *p* < 0.01, *** *p* < 0.001, Student’s *t*-test.

**Figure 2 ijms-24-12632-f002:**
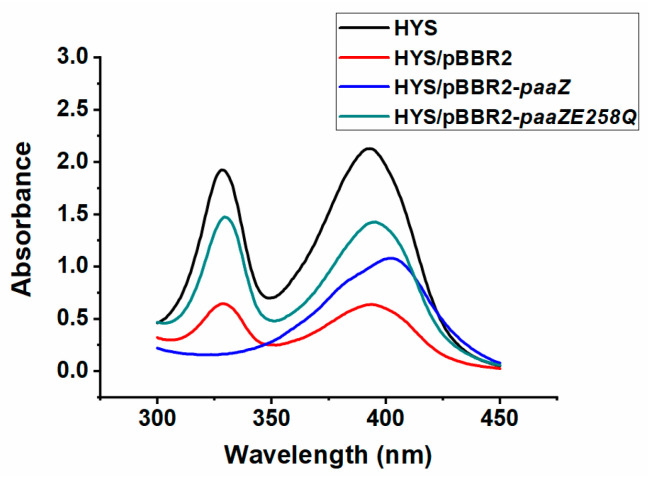
Effects of *orf13* (*paaZ*) on the production of 7-HT in *P. donghuensis* HYS. Absorption spectra of the filtered supernatants of 24 h MKB cultures from wild-type HYS, HYS/pBBR2, HYS/pBBR2-*paaZ*, and HYS/pBBR2-*paaZ*E258Q mutant strains.

**Figure 3 ijms-24-12632-f003:**
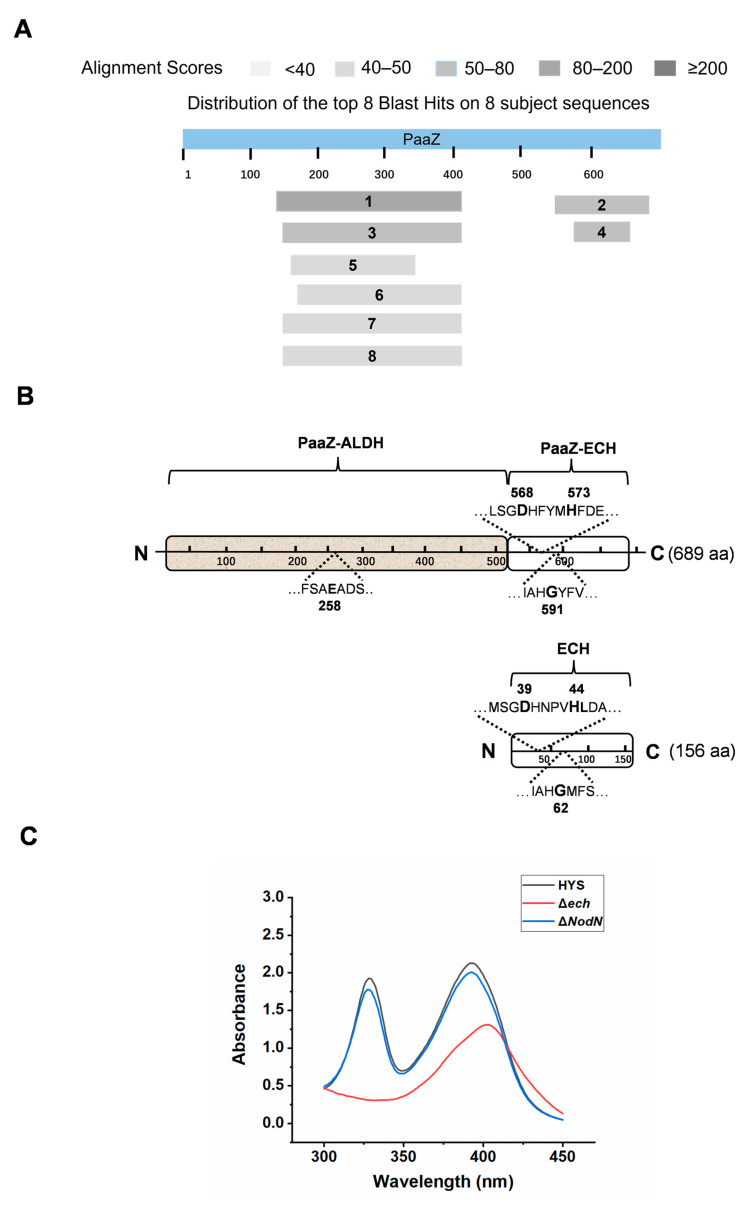
Alignment and analysis of *orf13* homologous genes obtained from a genome-wide BLASTP search in *P. donghuensis* HYS and their relationship with 7-HT biosynthesis. (**A**) Distribution of the eight proteins with homology to the N-terminal ALDH domain or the C-terminal ECH domain of ORF13 (PaaZ) obtained by the whole genome alignment search of *P. donghuensis* HYS. (**B**) Schematic view of the two proteins ORF13 (PaaZ) and UW3_RS0113785 (ECH) from *P. donghuensis* HYS. The N-terminal ALDH domain and C-terminal ECH domain of ORF13 (PaaZ) and UW3_RS0113785 (ECH) are displayed separately. The amino acids highlighted in bold are conserved catalytic sites in the domains, and the numbers are their positions in the protein. ORF13 (PaaZ), N-terminal: Glu-258. ORF13 (PaaZ), C-terminal: Asp-568, His-573, Gly-591. UW3_RS0113785 (ECH): Asp-39, His-44, Gly-62. (**C**) Absorption spectra of the filtered supernatants of 24 h MKB cultures from wild-type HYS, Δ*ech*, and Δ*NodN* mutant strains.

**Figure 4 ijms-24-12632-f004:**
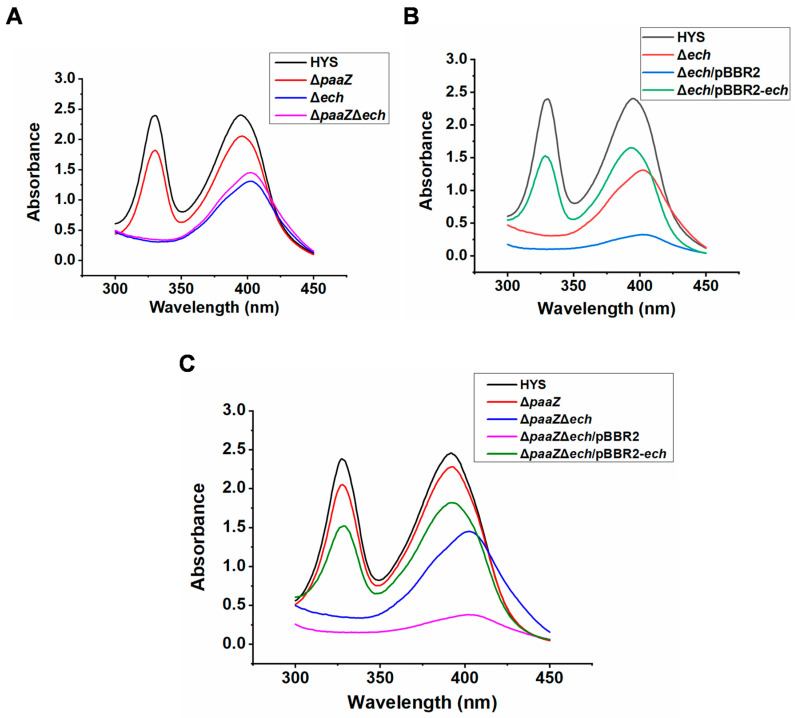
Different effects of the *paaZ* and *ech* genes on the production of 7-HT in *P. donghuensis* HYS. (**A**) Absorption spectra of the filtered supernatants of 24 h MKB cultures from wild-type HYS, Δ*paaZ*, Δ*ech*, and Δ*paaZ*Δ*ech* strains. (**B**) Absorption spectra of the filtered supernatants of 24 h MKB cultures from wild-type HYS, the Δ*ech*-deleted strain, the Δ*ech*/pBBR2 strain, and the Δ*ech*/pBBR2-*ech* strain. (**C**) Absorption spectra of the filtered supernatants of 24 h MKB cultures from wild-type HYS, Δ*paaZ*, Δ*paaZ*Δ*ech*, Δ*paaZ*Δ*ech*/pBBR2, and Δ*paaZ*Δ*ech*/pBBR2-*ech*-derivative strains.

**Figure 5 ijms-24-12632-f005:**
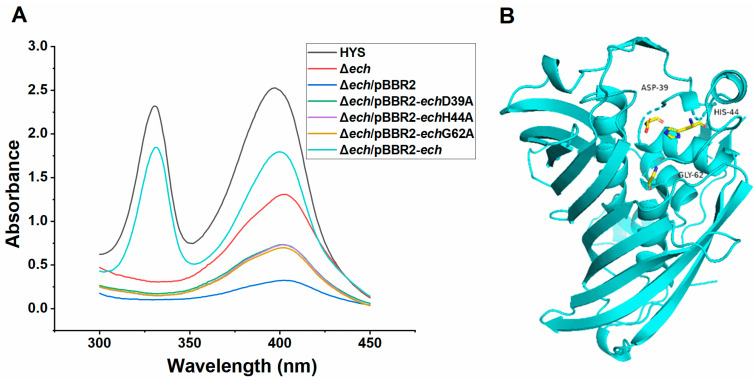
Critical residues of *ech* control with the involvement of *ech* in 7-HT biosynthesis and the effect of D39A, H44A, and G62A point mutations of these residues on 7-HT production in *P. donghuensis* HYS. (**A**) Absorption spectra of the filtered supernatants of 24 h MKB cultures from wild-type HYS, Δ*ech*, Δ*ech*/pBBR2, Δ*ech*/pBBR2-*ech*D39A, Δ*ech*/pBBR2-*ech*H44A, Δ*ech*/pBBR2-*ech*G62A, and Δ*ech*/pBBR2-*ech* strains. (**B**) The predicted 3D (three-dimensional) spatial protein structure for the *ech* of *P. donghuensis* HYS. The key residues Asp39, His44, and Gly62 are represented as a stick model, which indicates the location of the active sites of the enzyme. The black font shows the position of key residues in the sequence. The figure was prepared with SWISS-MODEL and PYMOL 2.5.2.

**Figure 6 ijms-24-12632-f006:**
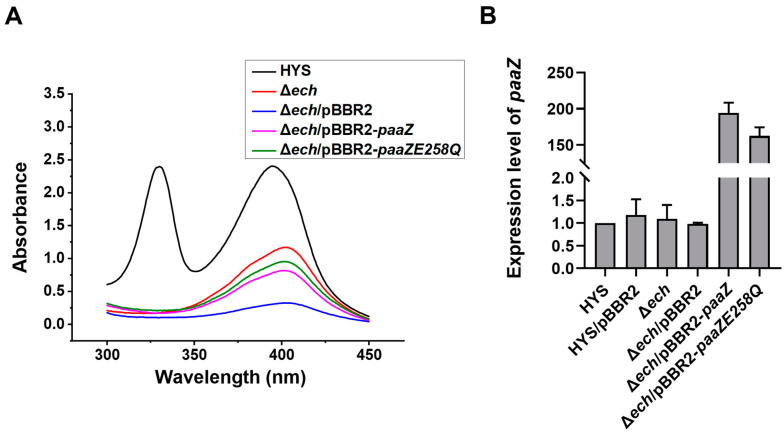
Δ*ech* introduced with *PaaZ* or *paaZ*E258Q of *P. donghuensis* HYS could not restore the production of 7-HT in the Δ*ech strain*. (**A**) Absorption spectra of the filtered supernatants of 24 h MKB cultures from wild-type HYS, Δ*ech*, Δ*ech*/pBBR2, *Δech pBBR2*-*paaZ*, and Δ*ech*/pBBR2-*paaZE258Q*-derivative strains. (**B**) The relative expressions of *orf13* (*paaZ*) in HYS/pBBR2, Δ*ech*, Δ*ech*/pBBR2, Δ*ech*/pBBR2-*paaZ*, and Δ*ech*/pBBR2-*paaZ*E258Q-derivative strains compared with that in the wild-type HYS strain was measured by qPCR and normalized using the rpoB gene. The error bars indicate standard deviations.

**Figure 7 ijms-24-12632-f007:**
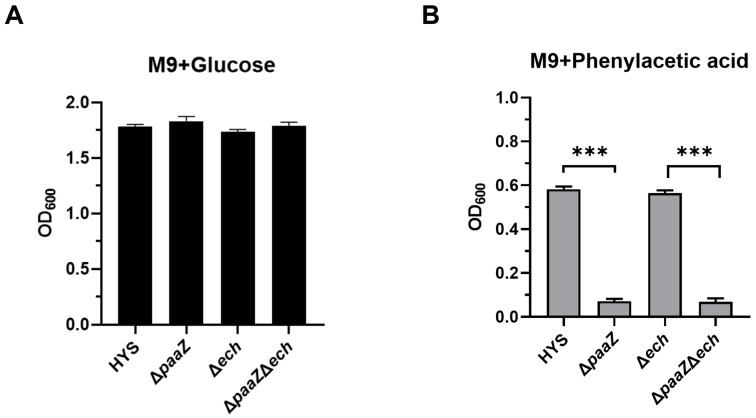
Sole carbon source tested for wild-type HYS, Δ*paaZ*, Δ*ech*, and Δ*paaZ*Δ*ech* mutant strains at the stationary phase. (**A**) Growth in M9 minimal medium with 0.4% glucose as a sole carbon source (black bar). (**B**) Growth in M9 minimal medium with 0.6 mg/mL PAA as a sole carbon source (grey bar). Each value is the average from three different cultures ± the standard deviation. The growth is expressed as OD_600_. OD_600_, optical density at 600 nm. *** *p* < 0.001, Student’s *t*-test.

**Figure 8 ijms-24-12632-f008:**
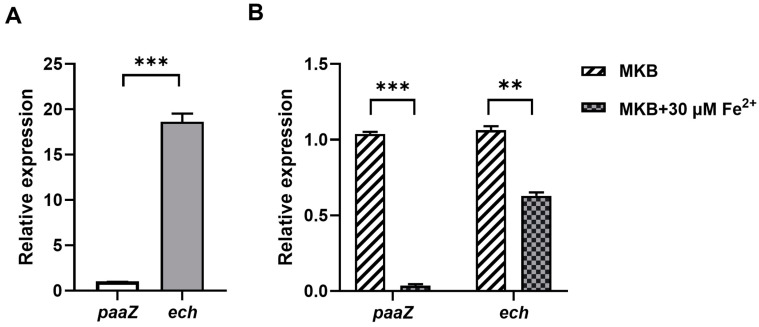
Relative expression levels of the *paaZ* and *ech* genes in the *P. donghuensis* HYS strain. (**A**) Relative transcriptional differences between the *paaZ* and *ech* genes in the exponential phase. Transcriptional levels are shown as the relative expression of the *paaZ* and *ech* compared to the expression of the *rpoB* gene at the exponential phase. (**B**) Effect of iron concentration on the expression of the *paaZ* and *ech* genes in the exponential phase. Quantitative fluorescence PCR was used to detect the transcription levels of *paaZ* and *ech* at MKB or MKB+ 30 μM FeSO_4_. Transcriptional levels are shown as the relative expression of the *paaZ* and *ech* compared to the expression of the *rpoB* gene at the exponential phase, as measured by qPCR. The error bars indicate the standard deviations (*n* = 3). ** *p* < 0.01, *** *p* < 0.001, Student’s *t*-test.

**Table 1 ijms-24-12632-t001:** Characteristics of PaaZ homologous protein ^a^ in *P. donghuensis* HYS.

Sequence Number	Description	Scientific Name	Max Score	Total Score	Query Cover	E Value	Per. ^b^ Ident	Acc. ^c^ Len	Accession ^d^
1	Aldehyde dehydrogenase family protein	*P. donghuensis* HYS	85.5	85.5	39%	7 × 10^−18^	29.60%	471	UW3_RS0102175
2	MaoC family dehydratase: (R)-hydratase [(R)-specific enoyl-CoA hydratase]	*P. donghuensis* HYS	62.8	62.8	19%	2 × 10^−16^	34.78%	156	UW3_RS0113785
3	NADP-dependent succinate-semialdehyde dehydrogenase	*P. donghuensis* HYS	62.4	62.4	37%	2 × 10^−10^	26.32%	480	UW3_RS0109265
4	MaoC family dehydratase: NodN (nodulation factor N)	*P. donghuensis* HYS	55.1	55.1	12%	2 × 10^−9^	32.18%	151	UW3_RS0112810
5	Aldehyde dehydrogenase family protein	*P. donghuensis* HYS	47.8	47.8	25%	6 × 10^−6^	25.56%	478	UW3_RS0116830
6	Aldehyde dehydrogenase family protein	*P. donghuensis* HYS	46.6	46.6	36%	1 × 10^−5^	22.64%	463	UW3_RS0100265
7	Aldehyde dehydrogenase family protein	*P. donghuensis* HYS	46.6	46.6	37%	1 × 10^−5^	24.09%	472	UW3_RS0100195
8	5-carboxymethyl-2-hydroxymuconate semialdehyde dehydrogenase	*P. donghuensis* HYS	44.3	44.3	37%	8 × 10^−5^	20.75%	486	UW3_RS0108160

^a^ Similarity values are for the most similar protein, determined by BLASTP analysis. ^b^ Percentage identity. ^c^ Amino acid length of the accession protein. ^d^ Gene id in *P. donghuensis* HYS.

## Data Availability

All data generated or analyzed during this study are included in this published article (and its Appendix A).

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
