# Peer review of "The Gene paaZ of the Phenylacetic Acid (PAA) Catabolic Pathway Branching Point and ech outside the PAA Catabolon Gene Cluster Are Synergistically Involved in the Biosynthesis of the Iron Scavenger 7-Hydroxytropolone in Pseudomonas donghuensis HYS"

_ijms, 2023, doi:10.3390/ijms241612632_

Round 1

Reviewer 1 Report

The article “The Gene paaZ of Phenylacetic Acid (PAA) Catabolic Pathway Branching Point and ech Outside PAA Catabolon Gene Cluster Are Synergistically Involved in the Biosynthesis of the Iron Scavenger 7-Hydroxytropolone in Pseudomonas donghuensis HYS” by Panning Wang, Yaqian Xiao, Donghao Gao, Yan Long and Zhixiong Xie focuses on the study of the iron scavenger 7-hydroxytropolone. It was secreted by Pseudomonas donghuensis HYS. 7-hydroxytropolone supports iron chelation and also contributes to the pathogenicity of P. donghuensis HYS to Caenorhabditis elegans. Furthermore, P. donghuensis HYS is more pathogenic to Caenorhabditis elegans than P. aeruginosa to C. elegans. The article contains 55 references, 32 of which are older than 2016. It is advisable to extend the list of references, or even better, to replace old references if possible. Overall, the article is very informative. The results section is very concise and clearly written. There are sufficient figures and tables, including supplementary material. However, the introduction and discussion could be improved.

It is recommended that the authors consider the following comments:

Lines 13-14 - this should be removed from the abstract as it is not the result of their work; and in lines 54-55 the authors write the same and make a reference.

Lines 103-124 - this should not be in the Introduction. This part is more like a summary or conclusion. This part should be removed and the aim of the paper (which is missing) should be given instead.

Lines 450-468 - These lines need improvement. They are a summary of the progress of the work and the results and do not contain any discussion. It is recommended to either shorten this part or to rewrite it better by comparing it with other tribes.

Moderate editing of English required

Reviewer 2 Report

Overall, I find the manuscript to be intriguing and informative, shedding light on the biosynthesis of the iron scavenger 7-hydroxytropolone (7-HT) in Pseudomonas donghuensis HYS. The authors have made a significant effort to investigate the role of phenylacetic acid (PAA) catabolon genes in cluster 2 and two specific genes, paaZ (orf13) and ech, in the production of 7-HT. However, some improvements are necessary to enhance clarity and focus. My specific comments are as follows:

  1. The abstract is informative but a bit lengthy. Consider shortening it to include only the main findings and their significance. Avoid describing specific findings from the manuscript in the abstract, and focus on summarizing the key results. Additionally, lines 11-15 should be revised for better coherence. From the first reading, it was not clear to me that you were describing the background, and not your own results.

  2. The introduction should be carefully revised to avoid repetition of information (for example, lines 54-55 and lines 73-75).
  3. Revise lines 103-124 and make them clearer.

  4.  While the introduction gives an adequate background on the topic, it lacks a clear statement of the main goal of the study. 

  5. The information in lines 507-522 seems more appropriate for the conclusion section rather than the main body of the manuscript. 

The manuscript could benefit from a language revision to enhance readability. Some sentences are convoluted and may be challenging for readers to follow.

Round 2

Reviewer 1 Report

The manuscript has been revised by the authors and is ready for publication.